# Laser frequency stabilization based on a universal sub-Doppler NICE-OHMS instrumentation for the potential application in atmospheric Lidar

Yueting Zhou[1,5], Jianxin Liu[1,5], Songjie Guo[1,5], Gang Zhao[1,5], Weiguang Ma[1,5], Zhensong Cao[2], Lei Dong[1,5], Lei Zhang[1,5], Wangbao Yin[1,5], Yongqian Wu[3], Lianxuan Xiao[1,5], Ove Axner[4], Suotang Jia[1,5]

[1]State Key Laboratory of Quantum Optics & Quantum Optics Devices, Institute of Laser Spectroscopy, Shanxi University, Taiyuan, 030006, China
[2]Key Laboratory of Atmospheric Composition and Radiation,Anhui Institute of Optics and Fine Mechanics, Chinese Academy of Sciences, Hefei, 230031, China
[3]Institute of Optics and Electronics, Chinese Academy of Sciences, 610209, Chengdu, China
[4]Department of Physics, Umeå University, SE-901 87 Umeå, Sweden
[5]Collaborative Innovation Center of Extreme Optics, Shanxi University, Taiyuan, 030006, China

*Correspondence to*: Weiguang Ma (mwg@sxu.edu.cn)

**Abstract.** Lidar is an effective tool for high altitude atmospheric measurement in which a weak absorption line for the target gas is selected to ensure a large optical depth. The laser frequency stabilization to the line center is required and a sub-Doppler (sD) spectroscopy of the target line is preferred as a frequency reference. In this paper, a novel universal sD NICE-OHMS instrumentation based on a fiber coupled optical single-sideband electro-optic-modulator (f-SSM) for the potential application in atmospheric Lidar for different target gases with different type of lasers is reported. The f-SSM can replace all frequency actuators in the system, so as to eliminate the individual design of feedback servos that often are tailored for each laser. The universality of the instrumentation was demonstrated by the alternative use of either an Er-doped fiber laser or a whispering gallery mode laser. Then the instruments based on both lasers were used to produce the sD signals of acetylene which worked as a frequency reference to stabilize the laser. By performing the lockings, relative frequency stabilizations of $8.3\times10^{-13}$ and $7.5\times10^{-13}$ at integration time of 240 s were demonstrated.

## 1 Introduction

The atmospheric Lidar can study the atmospheric properties from the ground up to the top of the atmosphere by using a laser. In order to get a large dynamic range in height, a relatively weak transition should be selected to ensure a large optical depth. Meanwhile, the laser frequency should be stabilized to the line center for a valid detection. In reality, the required frequency stability is ranging from hundreds of kHz to dozens of MHz, which relies on the linewidth of the target gas spectrum (Ehret et al., 2008). Up to now, the error for frequency stabilization in most of the Lidar system is measured by a high precision wavelength meter, which strongly depends on the stability of the wavelength meter. A local absorption spectroscopy of target gas is another alternative reference for active frequency stabilization. Therefore, the Jet Propulsion Laboratory in USA

has designed a frequency stabilization system for coherent Lidar application with frequency error of 2.06 MHz for detection of $v_1+v_3$ band of acetylene (Meras et al., 2008); The group in University of Maryland has designed another frequency stabilization system with frequency drift better than 0.3 MHz for the detection of $CO_2$ at the wavelength of 1572 nm (Numata et al., 2011). Benefiting from the narrow width of sub-Doppler (sD) absorption spectroscopy, it is the most suitable reference to get a high quality frequency stabilization.

sD spectroscopy is also an important tool for a number of other scientific research fields, including assessments of molecular spectra, investigation of chemical kinetics, realization of frequency standards, and assessment of fundamental constants(Moretti et al., 2013; Parthey et al., 2011; Tarallo et al., 2011; Taubman et al., 2004). sD spectroscopy provides responses with high spectral resolution that are free from Doppler-broadening (Db). To obtain such signals, high laser powers and low pressure conditions are normally used. To circumvent the use of such lasers, sD spectroscopy can be performed in Fabry-Perot (FP) cavities, since such can provide intracavity powers that are significantly higher than that of the impinging light field (by a factor of $F/\pi$, where F is the finesses of the cavity)(Delabachelerie et al., 1994; Ishibashi and Sasada, 1999). Since high-finesse FP cavities have narrow transmission modes, for most types of applications, the frequency of the laser field needs to be locked to that of the cavity mode, often by use of the Pound-Drever-Hall (PDH) technique(Drewer et al., 1983).

A drawback when absorption spectroscopy (AS) is combined with high finesse cavities is that, in its simplest realization, i.e. when cavity enhanced direct absorption spectrometry is performed, it is susceptible to frequency-to-amplitude converted noise caused by the residual frequency jitter between the frequencies of the laser field and the cavity mode that deteriorates the signal (Ye et al., 1996). As a means to circumvent this, the technique of noise-immune cavity-enhanced optical heterodyne molecular spectroscopy (NICE-OHMS) was invented. In this, the laser light is modulated at a radio-frequency (RF) to perform frequency modulation spectroscopy (FMS). By locking the modulation frequency of the FMS to the free spectral range (FSR) of the cavity, NICE-OHMS does not only benefit from the ability of FMS to reduce 1/f noise, which often is the dominating type of noise of the laser intensity, it also provides immunity to the frequency-to-amplitude noise(Axner et al., 2014; Ma et al., 1999; Ma et al., 2016; Ye et al., 1996). It has therefore large potential for a variety of applications.

NICE-OHMS was first realized by Jun Ye et al. in 1996 for frequency reference purpose(Ye et al., 1996). By use of an extremely well stabilized fixed-frequency Nd:YAG laser and a cavity with a finesse of $10^5$, a $C_2HD$ transition at 1064 nm was detected by use of a sD response. This first realization of NICE-OHMS demonstrated an unprecedented detection sensitivity of $1\times10^{-14}$ cm$^{-1}$ at 1 s averaging time.

A variety of tunable lasers, including external cavity diode laser (ECDL) (Bell et al., 2009; Chen and Liu, 2015; Dinesan et al., 2014; Ishibashi and Sasada, 2000; Saraf et al., 2016; van Leeuwen and Wilson, 2004), quantum cascade laser (QCL) (Taubman et al., 2004), optical parametric oscillator (OPO)(Hausmaninger et al., 2015; Talicska et al., 2016; Taubman et al., 2004), quantum-dot laser (Chen and Liu, 2017), distributed feedback laser (DFB) (Foltynowicz et al., 2010) and whispering gallery mode (WGM) laser (Zhao et al., 2017), have been applied to the NICE-OHMS system over the years. The most

persistent development of NICE-OHMS has been performed based on erbium-doped fiber lasers (EDFL), which lase in the near-IR (NIR) region (Ehlers et al., 2012; Schmidt et al., 2007). This has opened up for the use of fiber-coupled optical components, e.g. fiber-coupled acoustic-optic modulators (f-AOM) to expand the bandwidth for PDH locking and fiber-coupled electro-optic-modulator (f-EOM) that can be driven by fairly low RF powers. This has simplified, to a large extent,

the experimental setup and improved on the applicability of NICE-OHMS.

Despite the fact that the NICE-OHMS technique has been stated as noise-immune, the immunity is adversely affected when a NICE-OHMS signal is detected, either that from the molecular species under scrutiny or that from the optical system. Under these conditions, the quality of the detected signal can be limited by the performance of the locking procedure (Ma et al., 2016; Schmidt et al., 2010). This implies that it is of importance to lock the frequency of the laser to that of the cavity as

tightly as possible. This requires both a sufficiently large locking bandwidth and a high gain at low frequencies. In order to provide the former, more than one frequency actuator is usually utilized often a piezoelectric transducer (PZT) inside the laser and an external AOM. In addition, to prevent oscillations that can take place close to resonances of the actuators, in particular those from PZT, individual designs of proportional integral differential (PID) servo, specifically tailored for each laser, need to be developed and realized. This complicates significantly the construction of NICE-OHMS system and impairs

its applicability (Ehlers, 2014; Foltynowicz, 2009).

In recent years, a commercial fiber coupled optical single-sideband electro-optic-modulator (f-SSM), sometimes in the literature also referred to as an optical single-sideband suppressed-carrier modulator (Loayssa et al., 2001), has been used to shift the laser frequency with considerable tuning range and large modulation bandwidth. Such a device is composed of two Mach-Zehnder interferometers, each driven by a RF. By adjusting the phase difference of the two RF signals it can be made

to output a single sideband (Gatti et al., 2015). As a universal frequency shifter, it can replace all other frequency actuators in NICE-OHMS that are used for controlling the frequency of the laser light. Therefore this can facilitate the realization of a universal NICE-OHMS system that can be used with a variety of lasers.

This paper presents a universal NICE-OHMS setup for sD spectroscopy based on an f-SSM (MU-SSB-N-15-PM-FCAPC, Beijing Keyang Photonics Co. Ltd, China) for the potential application in the atmospheric Lidar. The universality of this

spectrometer is demonstrated by the alternative use of an EDFL (Adjustik E15 PztS PM, NKT Photonics, Denmark) and a WGM laser (OE4020-153806-PA02, OEwaves, USA). The applicability of the instrumentation was demonstrated by detection of sD NICE-OHMS signals of acetylene from both lasers, at 1531 and 1538 nm. They were then worked as references to stabilize the laser output frequencies to the target acetylene transitions. From 2014 to 2017, NICE-OHMS technique based frequency stabilizations has been done by several groups, the relative frequency stabilities of $5 \times 10^{-13}$

(Dinesan et al., 2014; Dinesan et al., 2015)and $6 \times 10^{-11}$ (Chen and Liu, 2017) have been demonstrated for an integration time of 1s, which are suitable for the future applications in the atmospheric Lidar.

## 2 Experimental setup and PDH locking performance

The experimental setup is shown in Fig. 1. The output of the laser passed through, in sequence, an f-SSM driven by a voltage-controlled oscillator (VCO1, SG386, Stanford Research Systems, USA), a fiber-coupled erbium doped fiber amplifier (f-EDFA) that is used to amplify and stabilize the laser power, and a f-EOM with proton exchanged waveguide (to suppress the residual amplitude modulation, RAM) (Silander et al., 2012) before it entered a fiber-coupled collimator (f-C), to be sent into free space. This implies that the laser frequency could be detuned by changing the frequency of the RF signal provided by the VCO1. Two modulation frequencies, 25.16 and 380.16 MHz were imposed on the laser light by the use of a single f-EOM for the PDH locking and the FMS, providing modulation indices of 0.19 and 0.8, respectively(Silander et al., 2012). Before impinging onto the cavity, the laser passed through a mode-matching lens (MML), a half wave plate ($\lambda/2$), a polarization beam splitter (PBS) and a quarter wave plate ($\lambda/4$). Both of the Lorenzian linewidths of the applied EDFL and WGM lasers are in the order of 100 Hz.

The cavity was composed by a plane and a concave mirror with a finesse of 2300 (estimated by cavity ringdown measurement) and cavity modes with full width at half maximum (FWHM) of 160 kHz. 52 % of the incident optical power was coupled into the cavity. The two mirrors were glued on two PZTs which, in turn, were glued to the cavity spacer, which was made of Zerodur. The mirror distance was 39.4 cm, which gave rise to a FSR of 380.16 MHz. During the measurement, the averaged temperature stability of the cavity is better than 0.5 K/hour since the cavity is exposed in the air.

The reflection of the cavity was deflected by the PBS to a photodetector (PD1) for the PDH and the DeVoe-Brewer (DVB) lockings (DeVoe and Brewer, 1984), the latter used to lock the modulation frequency of the FMS to the FSR of the cavity. The light transmitted through the cavity was focused onto a photodetector (PD2), which provided the NICE-OHMS signal. To minimize the amount of background signals from etalons, the optical components were, as much as possible, tilted and placed at etalon immune distance (EID) (Ehlers et al., 2014).

To avoid the complex design of bandpass filter working in the RF frequency region, a 25.16 MHz signal for the PDH locking was created from a low pass filtered beat signal between two RF frequencies (at 380.16 and 355 MHz) that were generated by two RF generators(Bell et al., 2009). The error signal for the PDH locking was obtained by demodulating the reflected light from the cavity at 25.16 MHz which then was sent to the VCO1 through a proportional–integral–derivative servo (PID1). The design of this servo was thereby only dictated by the frequency response of the VCO1 and not by those of the lasers. To provide high gain in the low frequency region, the locking servo contained two integrators.

It should be noted that although both aforementioned lasers can be detuned by a PZT inside the laser, the two PZTs have different frequency responses; low-pass frequency corners at 4.5 and 50 kHz, and oscillation frequencies at 34 and 48 kHz for the EDFL, and at 38 kHz for the WGM laser, respectively. Therefore, in conventional setups, careful designs of locking servos (with individually optimized parameter values) and an external f-AOM would have been needed (the latter to expand the locking bandwidth, preferably up to around 100 kHz). In our system, on the other hand, the employment of an f-SSM as

a unique and fast frequency shifter can eliminate the individually designed PDH servos that are needed for different types of lasers.

Figure 2 shows the frequency noise spectra of the in-loop error signals for PDH locking, which is measured by a electronical spectrum analyzer (FSW, Rohde&Schwarz, Germany). The black and red lines represent the situations for the EDFL and the WGM lasers, respectively, measured without any change of servo parameters for the PDH locking. Both the Video Bandwidth (VBW) and Resolution Bandwidth (RBW) in the measurement are 30Hz. It can be seen that both spectra have a similar shape and that the servo reduces the noise in the low frequency region efficiently due to the use of two integrators in series. The peaks at 19 and 38 kHz come from the common ground in the lab which has been found difficult to eliminate, while the bumps at 200 kHz indicate the bandwidth of the PDH servo (Fox et al., 2002). The difference between the two curves is attributed to dissimilar phase noises of the two lasers. This demonstrates that the locking bandwidth is 200 kHz irrespective of which laser the NICE-OHMS system is utilized.

It is worth to note that the PDH locking bandwidth is mainly limited by the VCO1, which in this work had a bandwidth of 100 kHz. As suggested by D. Gatti, et al., the bandwidth of PDH servo can be increased to 5 MHz if a higher bandwidth VCO and a good PID servo would be employed (Gatti et al., 2015). Such an action will expand the applicability of the NICE-OHMS system to lasers with larger linewidth, e.g. DFB and ECDL.

## 3 Measurement of sD NICE-OHMS signals

Two acetylene transitions, one at 1531 nm and another at 1538 nm, were addressed by the two lasers while sD signals were monitored. Figure 3 shows, by the black and red curves, two sD NICE-OHMS signals, from 100 mTorr total pressure and 1000 ppm concentration of $C_2H_2$, detected at dispersion phase by the EDFL and WGM lasers, respectively. The pressure leaking rate for the cavity was smaller than 0.01 mTorr/min after it was closed by the vacuum valve (Leycon 215379, Oerlikon, Germany).

The EDFL addressed the $P_{10}(e)$ transition of the $v_1+v_3$ band of $C_2H_2$ at 1530.976 nm, which has a line strength of $4.0\times10^{-21}$ $cm^{-1}/(mol^{-1}cm^{-2})$ and a dipole moment of 8.24 mD. The saturation power under the pressure of 100 mTorr was around 700 mW (Foltynowicz et al., 2008b). The FWHM of the sD NICE-OHMS signal was about 3.5 MHz. It can be noted that by assuming that the transit time broadening is 188 kHz and the degree of saturation is close to 6 (for an incident power of 5.86 mW), it can be estimated that the pressure broadening should be around 1.05 MHz. This is inconsistent with the calculated broadening of 0.64 MHz from the pressure broadening coefficient provided by the HITRAN database (a similar disagreement has previous been reported (Foltynowicz et al., 2008a)). The frequency scan range, which is 16 MHz, is limited by the performance of VCO1 (whose resolution is proportional to the total scanning range). The degree of saturation was estimated from the ratio of amplitudes of sD and Db signals (Axner et al., 2008).

The WGM laser addressed the $P_{21}(e)$ transition of the $v_1+v_3$ band of $C_2H_2$ at 1538.058 nm, which has a line strength of $3.4\times10^{-21}$ $cm^{-1}/(mol^{-1}cm^{-2})$ and a dipole moment of 7.94 mD. The corresponding saturation power under the pressure of 100

mTorr was around 800mW. The FWHM of the sD NICE-OHMS signal was about 3.8 MHz. By assuming both the similar transit time broadening and degree of saturation as above (for an incident power of 5.92 mW), it can be estimated that the pressure broadening of this transition should be around 1.15 MHz.

The signal-to-noise ratios for the two sD signals, in figure 3, were similar, assessed to 380, indicating detection sensitivities of $1.0\times10^{-9}$ and $0.9\times10^{-9}$ cm$^{-1}$, respectively. The sensitivities are around four orders of magnitude worse than the best results in the field (Zhao et al., 2018), the reasons for which are the adopted lower finesse cavity, not fully optimised optical system and the different evaluation method of sensitivity (by evaluating the Allan variance of the fitted amplitude of long-term background measurements by a NICE-OHMS lineshape in all other EDFL based NICE-OHMS papers). The blue lines in Figure 3 show the background NICE-OHMS signals measured from an empty cavity that originates from etalons generated between optical components.

## 4 Laser locking to the sD NICE-OHMS signal

The frequency of each laser was then locked to the center of the addressed transition by feeding the sD signal back to the cavity PZT. The performance of the this frequency stabilization was assessed by Allan plot of the frequency deviation estimated from the error signals, i.e. the sD NICE-OHMS signals, calibrated by the slopes at the zero crossing point of the sD signals. The two upper-most panels in Figure 4 show the frequency deviations for the EDFL and the WGM lasers, respectively, measured over 4000 s with a sampling rate of 30 Hz. It can be seen that both frequency deviations jitter around zero with little drift, and that the deviations for the WGM laser are larger than those for the EDFL, consistent with the noise spectra shown in Figure 2.

The lower panels in Figure 4 shows that for integration times up to 100 sec, the deviations in frequency have a white noise response, as is indicated by the dotted lines, which show a $\tau^{-1/2}$ dependence. The Allan plots furthermore show that, for an integration time of 1s, the relative frequency deviations of the of the EDFL (black dot line) and the WGM lasers (red dot line) are $9.4\times10^{-12}$ and $1.3\times10^{-11}$, respectively; however, for an integration time of 240 sec, the relative frequency deviations are $8.3\times10^{-13}$ and $7.5\times10^{-13}$, respectively (which corresponds to frequency stabilities of 0.16 kHz and 0.15 kHz). The upturn of Allan deviation is due to the not fully suppressed residual amplitude modulation of fiber EOM, the not perfect design of PDH servo and etalon noise in the beam path.

However the laser frequency locking performance is strongly influenced by the amplitude and linewidth of sD NICE-OHMS signal which will be changed by the gas pressure and intracavtiy power. Although the locking results in the paper have satisfied the requirements of atmospheric LIDAR applications, if the NICE-OHMS system is optimized, an even better result can be expected.

## 5 Conclusion

We have demonstrated the construction of a universal NICE-OHMS instrumentation for sD spectroscopy for the potential application in atmospheric Lidar that is based on a single external frequency actuator, a fiber-coupled optical single-sideband electro-optic-modulator (f-SSM), to control and detune the laser frequency. The universality of the instrumentation has been tested by alternatively incorporating it to an EDFL and a WGM laser. It has been demonstrated that the same locking system, with a given PID servo, could be used to lock both lasers to a given cavity, with the same bandwidth (200 kHz). The use of a single external frequency actuator can eliminate the individual design of servos for each laser, which normally need to be specifically tailored for each laser. It also lowers the requirement of laser tunability for the spectrum measurement. It therefore significantly expands the applicability of the NICE-OHMS technique. Two $C_2H_2$ transitions were addressed by sD spectroscopy where the laser frequency was tuned by alteration of the radio frequency to the f-SSM. Finally, the two laser frequencies were locked to the transition centers of the sD NICE-OHMS features, resulting in relative frequency stabilizations of $8.3\times10^{-13}$ and $7.5\times10^{-13}$ for an integration time of 240 sec. Although the frequency stability can satisfy the requirements of a real Lidar system, the whole setup should be miniaturized and a seismic design is necessary before it is applied to the real field. On the other hand, if this setup is applied to the detection of $CO_2$, $CH_4$ or other gases, the saturation power of the target molecular transition, as a function of dipole moment, laser beam waist and total pressure, should be determined firstly to choose a suitable laser and a proper cavity design. The transition dipole moment is the function of its linestrength, life time and partition function (Ma et al., 2008). Normally, for a weak absorption line with the saturation power in the order of hundreds of watts, a cavity with finesse of more than tens thousands and a laser with its linewidth of the order of kHz are required., a cavity with finesse of more than tens thousands and a laser with its linewidth of the order of kHz are required. Anyway, the design of robust and universal sD NICE-OHMS instrumentation opens up a new application in atmospheric Lidar.

## 6 Acknowledgements

The work was supported by National Key R&D Program of China (Grant No. 2017YFA0304203), Changjiang Scholars and Innovative Research Team in University of Ministry of Education of China (Grant No. IRT_17R70), the Fund for Shanxi "1331 Project" Key Subjects Construction, 111 project (Grant No. D18001), the National Natural Science Foundation of China (Grant Nos. 61675122, 61875107, 61875108, 11704236, 61475093 and 61775125), the Research Project Supported by Shanxi Scholarship Council of China (2017-016), the Program for the Outstanding Innovative Teams of Higher Learning Institutions of Shanxi and Open Research fund of Key Laboratory of Atmospheric Optics, Chinese Academy of Sciences (JJ-2018-02). Ove Axner would also like to acknowledge support from the Swedish Research Foundation (*swe.* Vetenskapsrådet, Grant No. 621-2015-04374).

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

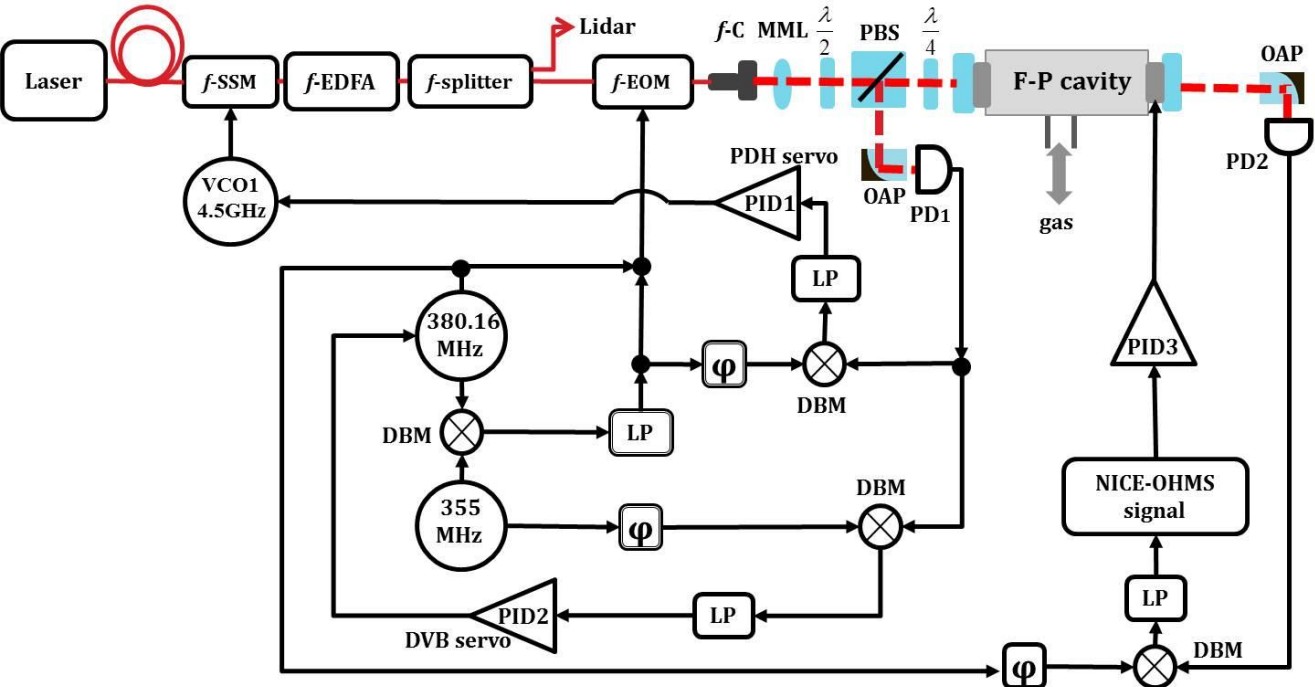

Figure 1: Experimental setup: f-SSM, fiber coupled single-sideband electronic optical modulator; f-EDFA, fiber coupled Erbium doped fiber amplifier; f-EOM, fiber coupled electronic optical phase modulator; f-C, fiber collimator; MML, mode-matching lens; $\lambda/2$, half wave plate; PBS, polarizing beam splitter; $\lambda/4$, quarter wave plate; OAP, off axis parabolic mirror; PD, photo detector; LP, Low pass filter; PID, Proportional–integral–derivative controller; VCO, voltage control oscillator; DBM, double balanced mixer; $\varphi$, phase shifter; DVB servo, DeVoe-Brewer Servo; F-P cavity, Fabry-Perot cavity.

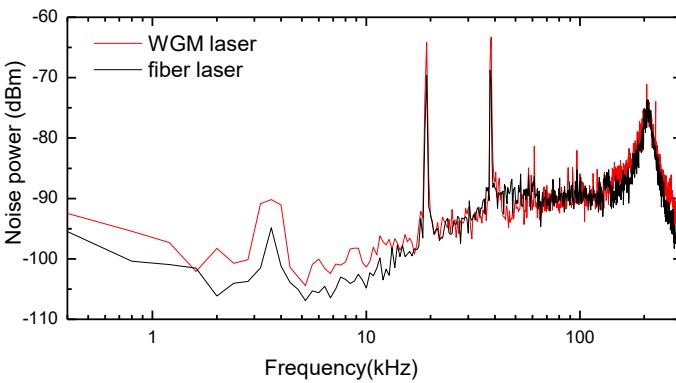

Figure 2: In loop frequency spectrum of the error signal of the PDH locking for EDFL (black curve) and WGM laser (red curve).

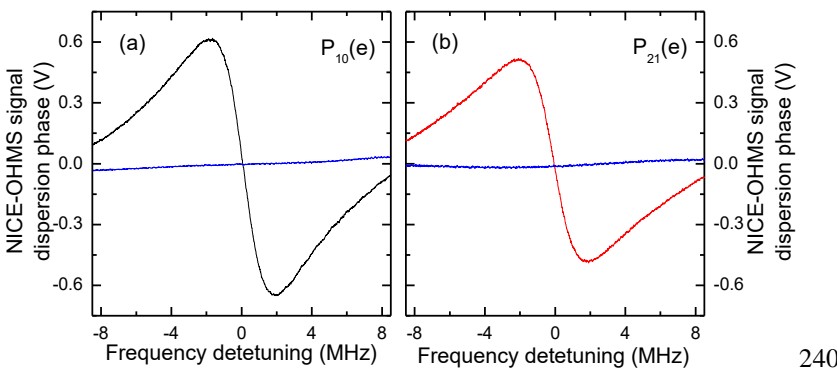

240

Figure 3: The sD NICE-OHMS signals detected at dispersion phase on the $P_{10}(e)$ and $P_{21}(e)$ transitions from 100 mTorr of acetylene detection by an EDFL and a WGM laser, respectively. The blue lines in both panels are the background signals when the cavity was evacuated.

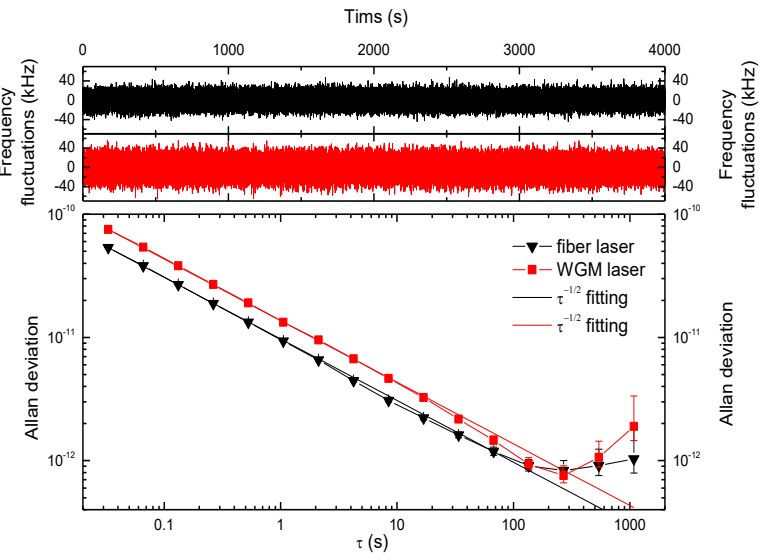

Figure 4: Upper two panels: the frequency deviation obtained from sD signals measured over 4000 s for the EDFL (black curve) and the
WGM laser (red curve); The lower panel: The Allan deviation of the frequency deviation for the EDFL (black curve) and the WGM laser
(red curve). The dotted lines correspond to the white noise response.