# Peer review of "Laser frequency stabilization based on a universal sub-Doppler NICE-OHMS instrumentation for the potential application in atmospheric Lidar"

_Atmospheric Measurement Techniques, 2018_

## Referee Comment (RC1) · Anonymous Referee #1 · 9 Jan 2019

Zhou et al present a system for laser stabilization based on NICE-OHMS. Instead of directly feeding back to the laser, they use a single-sideband modulator as a frequency actuator. This increases the flexibility of the system, as demonstrated by the similar performance achieved with two different lasers. While the system is well designed and implemented, I'm not sure that the results presented relevant enough to the atmospheric community or novel enough for publication in AMT.

For the applicability to the atmospheric community, more details need to be provided about the potential application to LIDAR:

- First, what frequency stability is required for LIDAR and why?

- My initial thought is that NICE-OHMS is overly complicated for this application. The motivation for using NICE-OHMS was not well justified. What performance has been achieved with a wavemeter and with other schemes such as WMS/FM spectroscopy in a cavity? Why is NICE-OHMS necessary?

- What gas is expected to be used for the LIDAR application ($C_2H_2$ is not the most atmospherically relevant)? How will the performance compare (i.e., what are the linestrengths, expected linewidths, etc.)?

For the novelty, the included Gatti et al reference has already demonstrated a wide bandwidth PDH lock using a single sideband modulator. This fact should be clarified in the current paper and the differences between this and the Gatti work should be discussed (e.g., the feedback bandwidth was higher in Gatti et al). Also, why was a lower bandwidth VCO used for VCO1? Demonstrating the ability to use a DFB or ECDL would make this a stronger paper.

In addition, before publication the language could use some editing.

Finally, I have some more specific comments/questions as well:

1. "DVB" and "F-P cavity" are not defined in the caption to Figure 1

2. Why were OAPs used instead of lenses?

3. For the LIDAR output, will the frequency need to be tuned, e.g., to do an on-line/off-line measurement? If so, how will this be accomplished?

4. For Figure 2, it would be better to give the noise PSD. Also, I assume that this is an in-loop measurement? This should be clarified. In addition, I believe that this should be corrected for the cavity low-pass filter effect (e.g., Fig 5 in Gatti et al)?

5. What causes the upturn in the Allan deviation past 200 seconds?

6. What was the temperature stability of the lab? How would the performance be affected by reduced temperature stability?

---

## Referee Comment (RC2) · Anonymous Referee #2 · 24 Jan 2019

General comments:

This paper develops a NICE-OHMS-based laser stabilisation system for potential application for lidar. The authors provide a brief overview of the relevant technical aspects of the paper, including sub-Doppler spectroscopy, the NICE-OHMS technique and the laser technology used in various NICE-OHMS demonstrations. The highlight of the paper is the application and use of a fiber-coupled optical single-sideband electo-optic modulator (f-SSM) to make the system useful for laser stabilisation to a variety of lasers, and they demonstrated this by stabilising both an Erbium-doped fiber laser (EDFL) and a whispering gallery mode (WGM) laser. After producing a sD NICE-

[Figure]

OHMS signal with each laser (using acetylene as a reference gas), each laser was then stabilised to that acetylene transition frequency using the sD NICE-OHMS signal produced. The stability of the stabilised output was measured and found to exceed the level needed for use in lidar systems.

The paper is well organised and clearly written, for the most part. Although the applicability of this system to lidar needs to be expanded upon, I see no reason why it could not be used as a stabilised seed laser for lidar.

Specific comments:

What was the acetylene gas source for your initial experimental realisation of this, and how did you measure that it was at 100mTorr? Do you know the long-term stability of the pressure in the system? Would you expect to use the same system setup "in the field" when supporting a lidar system?

Can you be more explicit about the stability requirements of a lidar system? What about the power required to stabilise such a system, can you give the details of the power requirement and can your system provide that power?

Lidar systems also would like to target other molecules, especially those related to greenhouse gas emissions ($CO_2$, methane, etc.) Can you comment on how adaptable your system is to these other molecular species?

Can you provide any comments at the end of your paper to indicate what changes would need to be made to your system for it to work reliably outside a research laboratory, "in the field"?

Page 3: Line 1: I am confused by this first sentence. The "noise immunity" obviously refers to "noise" sources - not the actual signal. The "background signal" you mention is induced by other processes unrelated to the "noise" in the "noise immunity" - from optical power variations in the incident light to the cavity, or coupling efficiency. Can you please clarify or remove this sentence?

Page 6: Line 6: "for integration times up to 240 sec" - I might argue that you have over-estimated your white noise response window, especially in the case of the fiber laser Allan deviation data (black points and curve). I would estimate (reading directly off Figure 4) that there is a white noise response out to 100-120 seconds (the intersection of your "white noise" line and a flat line indicating the ADEV value when the ADEV begins drifting up), rather than 240 seconds, for the fiber laser. By 240 seconds you are clearly out of the white noise regime. However, it is a bit difficult to see this clearly from the graph. Additionally, it is not obvious that there are any "dotted" lines on this plot, as mentioned in the figure caption - the "white noise response" lines appear solid.

Figure 4: To be able to better compare the signal-to-noise-ratio of the sD NICE-OHMS signals in sub-plots (a) and (b), it would be helpful to have the y-axes have the same minimum and maximum values (perhaps 0.7 or 0.65).

Technical corrections: Page 1: Line 1: Typically one would use "a universal" rather than "an universal". Line 26: "a relative weak" -> a relatively weak

Page 2: Line 22: "cavity with a finesse of 105" - > "cavity with a finesse of 100,000" or "cavity with a finesse of 10ˆ5" Line 24: should be rewritten - "sensitivity of 1x10ˆ-14 cm-1 at 1-s averaging time."

Page 3: Line 6/7: "more than one frequency actuators are usually utilized often a" -> "more than one frequency actuator is usually utilized, often a"

Page 5: Line 30: "The performance of the this frequency stabilization were assessed by" -> "The performance of the this frequency stabilization were was assessed by"

Page 6: Line 15: "testified" - > "tested"

Generally there is sometimes an issue of spacing between text and the beginning or end of parentheses that should be checked for throughout the paper.

It is a bit confusing to have two very similar References (Ehlers, 2014) and (Ehlers et al., 2014) in the paper text, there a way to differentiate? (Ehlers [Thesis], 2014)? Also,

I am not sure that the page 7, line 24 Ehlers reference, is correct. Perhaps use "PhD dissertation" rather than "Doctor". "Ehlers, P.: Further development of NICE-OHMS – an ultra-sensitive frequency-modulated cavity-enhanced laser-based spectroscopic technique for detection of molecules in gas phase, PhD dissertation, Umeå universitet, Umeå, 2014."

———————————————————

---

## Author Comment (AC2) · 26 Jan 2019

The answers to the reviewers' comments are as follows, the revised paper is added to the Supplement.

This paper develops a NICE-OHMS-based laser stabilisation system for potential application for lidar. The authors provide a brief overview of the relevant technical aspects of the paper, including sub-Doppler spectroscopy, the NICE-OHMS technique and the laser technology used in various NICE-OHMS demonstrations. The highlight of the paper is the application and use of a fiber-coupled optical single-sideband electooptic modulator (f-SSM) to make the system useful for laser stabilisation to a variety

of lasers, and they demonstrated this by stabilising both an Erbium-doped fiber laser (EDFL) and a whispering gallery mode (WGM) laser. After producing a sD NICEC1 OHMS signal with each laser (using acetylene as a reference gas), each laser was then stabilised to that acetylene transition frequency using the sD NICE-OHMS signal produced. The stability of the stabilised output was measured and found to exceed the level needed for use in lidar systems. The paper is well organised and clearly written, for the most part. Although the applicability of this system to lidar needs to be expanded upon, I see no reason why it could not be used as a stabilised seed laser for lidar.

Answer: Thanks for your positive comments.

Specific comments: What was the acetylene gas source for your initial experimental realization of this, and how did you measure that it was at 100mTorr? Do you know the long-term stability of the pressure in the system? Would you expect to use the same system setup "in the field" when supporting a Lidar system?

Answer: Thanks for the comments. "total pressure and 1000 ppm concentration" and "The pressure leaking rate for the cavity was smaller than 0.01 mTorr/min after it was closed by the vacuum valve (Leycon 215379, Oerlikon, Germany)." are inserted into the first paragraph of section 3. The sub-Doppler NICE-OHMS setup should be integrated and miniaturized before it is applied to the Lidar system. In this status, it can not be applied to the Lidar system.

Can you be more explicit about the stability requirements of a lidar system? What about the power required to stabilize such a system, can you give the details of the power requirement and can your system provide that power?

Answer: Thanks for the comments. "In reality, the required frequency stability is ranging from hundreds of kHz to dozens of MHz, which relies on the linewidth of the target gas spectrum [1]." and "Therefore, the Jet Propulsion Laboratory in USA has designed a frequency stabilization system for coherent Lidar application with frequency error

of 2.06 MHz for detection of $\nu 1+\nu 3$ band of acetylene [2]; The group in University of Maryland has designed another frequency stabilization system with frequency drift better than 0.3 MHz for the detection of CO2 at the wavelength of 1572 nm [3]." are insert to the first paragraph of section 1.

Lidar systems also would like to target other molecules, especially those related to greenhouse gas emissions (CO2, methane, etc.) Can you comment on how adaptable your system is to these other molecular species?

Answer: hanks for the comments. "However if this setup is applied to the detection of CO2, CH4 or other gases, the saturation power of the target molecular transition, as a function of dipole moment, laser beam waist and total pressure, should be determined firstly to choose a suitable laser and a proper cavity design. The transition dipole moment is the function of its linestrength, life time and partition function (Ma et al., 2008). Normally, for a weak absorption line with the saturation power in the order of hundreds of watts, a cavity with finesse of more than tens thousands and a laser with its linewidth of the order of kHz are required." is added to the last paragraph of section 5. The saturation powers for the target two transitions are also added in the proper positions in section 3.

Can you provide any comments at the end of your paper to indicate what changes would need to be made to your system for it to work reliably outside a research laboratory, "in the field"? Answer: Thanks for the comments. "Although the frequency stability can satisfy the requirements of a real Lidar system, the whole setup should be miniaturized and a seismic design is necessary before it is applied to the real field." is insert into the last paragraph of section 5.

Page 3: Line 1: I am confused by this first sentence. The "noise immunity" obviously refers to "noise" sources - not the actual signal. The "background signal" you mention is induced by other processes unrelated to the "noise" in the "noise immunity" – from optical power variations in the incident light to the cavity, or coupling efficiency. Can

you please clarify or remove this sentence?

Answer: Thanks for the comments. The noise in "noise immunity" refers solely to the frequency to amplitude noise. The influence of this noise to the NICE-OHMS signal (i.e. the demodulated cavity enhanced frequency modulation signal) is null only under the condition of empty cavity, zero residual amplitude modulation from EOM and other background noise before the cavity. If the laser addresses the molecular transition, the absorption will make the cavity transmitted triplet different noise level and therefore destroy the noise immunity property to some extend. We remove "often referred to as a background signal" from the line in the text.

Page 6: Line 6: "for integration times up to 240 sec" - I might argue that you have overestimated your white noise response window, especially in the case of the fiber laser Allan deviation data (black points and curve). I would estimate (reading directly off Figure 4) that there is a white noise response out to 100-120 seconds (the intersection of your "white noise" line and a flat line indicating the ADEV value when the ADEV begins drifting up), rather than 240 seconds, for the fiber laser. By 240 seconds you are clearly out of the white noise regime. However, it is a bit difficult to see this clearly from the graph. Additionally, it is not obvious that there are any "dotted" lines on this plot, as mentioned in the figure caption - the "white noise response" lines appear solid. Figure 3: To be able to better compare the signal-to-noise-ratio of the sD NICE-OHMS signals in sub-plots (a) and (b), it would be helpful to have the y-axes have the same minimum and maximum values (perhaps 0.7 or 0.65).

Answer: Thanks for your reminding. The corresponding changes have done in the revised version.

Technical corrections: Page 1: Line 1: Typically one would use "a universal" rather than "an universal". Line 26: "a relative weak" -> a relatively weak; Page 2: Line 22: "cavity with a finesse of 105" - > "cavity with a finesse of 100,000" or "cavity with a finesse of 10ËĘ5" Line 24: should be rewritten - "sensitivity of 1x10ËĘ-14 cm-1 at 1-s

averaging time." Page 3: Line 6/7: "more than one frequency actuators are usually utilized often a" ->"more than one frequency actuator is usually utilized, often a" Page 5: Line 30: "The performance of the this frequency stabilization were assessed by" -> "The performance of the this frequency stabilization were was assessed by" Page 6: Line 15: "testified" - > "tested"

Answer: The corrections are done in the revised version.

Generally there is sometimes an issue of spacing between text and the beginning or end of parentheses that should be checked for throughout the paper.

Answer: The corrections are done in the revised version.

It is a bit confusing to have two very similar References (Ehlers, 2014) and (Ehlers et al., 2014) in the paper text, there a way to differentiate? (Ehlers [Thesis], 2014)? Also, I am not sure that the page 7, line 24 Ehlers reference, is correct. Perhaps use "PhD dissertation" rather than "Doctor". "Ehlers, P.: Further development of NICE-OHMS – an ultra-sensitive frequency-modulated cavity-enhanced laser-based spectroscopic technique for detection of molecules in gas phase, PhD dissertation, Umeå universitet, Umeå, 2014."

Answer: The corrections for all the references are done in the revised version.

New added references

1. G. Ehret, C. Kiemle, M. Wirth, A. Amediek, A. Fix, and S. Houweling, "Space-borne remote sensing of $CO_2$, $CH_4$, and $N_2O$ by integrated path differential absorption lidar: a sensitivity analysis," Appl. Phys. B. 90, 593-608 (2008).

2. P. J. Meras, I. Y. Poberezhskiy, D. H. Chang, J. Levin, and G. D. Spiers, "Laser frequency stabilization for coherent lidar applications using novel all-fiber gas reference cell fabrication technique," in 24th International Laser Radar Conference, (Boulder, Colorado, 2008).

3. K. Numata, J. R. Chen, S. T. Wu, J. B. Abshire, and M. A. Krainak, "Frequency stabilization of distributed-feedback laser diodes at 1572 nm for lidar measurements of atmospheric carbon dioxide," Appl. Optics. 50, 1047-1056 (2011).

Please also note the supplement to this comment:
https://www.atmos-meas-tech-discuss.net/amt-2018-389/amt-2018-389-AC2-supplement.pdf

─────────────────────

[Figure]

**Supplement:**

**Laser frequency stabilization based on a universal sub-Doppler NICE-OHMS instrumentation for the potential application in atmospheric Lidar**

Yueting Zhou1,5, Jianxin Liu1,5, Songjie Guo1,5, Gang Zhao1,5, Weiguang Ma1,5, Zhensong Cao2, Lei 5 Dong1,5, Lei Zhang1,5, Wangbao Yin1,5, Yongqian Wu3, Lianxuan Xiao1,5, Ove Axner4, Suotang Jia1,5

[revised manuscript text omitted]

---

## Referee Comment (RC3) · Anonymous Referee #3 · 15 Feb 2019

The manuscript tilted by "Laser frequency stabilization based on an universal sub-Doppler NICE-OHMS instrumentation for the potential application in atmospheric Lidar" made a frequency stabilized laser based on cavity enhanced optical heterodyne molecular spectroscopy named as NICE-OHMS.

The text line of 30 in the page 5 is as following: "The performance of the frequency stabilization were assessed by Allan plot of the frequency deviation estimated from the error signals, i.e. the sD NICE-OHMS signals, calibrated by the slopes at the zero crossing point of the sD signals."

Analysis of servo error signal is not right way to characterize the performance of frequency stabilized laser. The electronic servo box can drive the laser frequency to make error signal at the zero-crossing point of the sD signals. Fig.3 shows the baseline of sD signals (blue line) is moving around the zero-crossing point. The servo box adjusted the laser frequency to make error signal at the zero-crossing point, therefore the laser frequency is unstable. So I hope the authors to make two independent frequency stabilized lasers and make analysis of the beating signal between two lasers to characterize the performance of frequency stabilized laser.

The performance of stabilized laser is dependent on the length of cavity, pressure of cell, input light power to the cavity, beam size inside of the cavity, cell temperature instability, the cavity output light power and the RAM in the sD signal. So the authors need to add analysis and optimization for the parameters which affect the frequency instability of the stabilized lasers.

The manuscript needs to be modified before publication.

---

## Author Response (AR1)

To The Editor of *Atmospheric Measurement Techniques*

Resubmission of manuscript amt-2018-389: "Laser frequency stabilization based on an universal sub-Doppler NICE-OHMS instrumentation for the potential application in atmospheric Lidar", by Yueting Zhou, Jianxing Liu, Songjie Guo, Gang Zhao, Weiguang Ma, Zhensong Cao, Lei Dong, Lei Zhang, Wangbao Yin, Yongqian Wu, Liantuan Xiao, Ove Axner, and Suotang Jia to *Atmospheric Measurement Techniques.*

Dear Editor,

We thank the three reviewers for their insightful and helpful comments. The comments have been scrutinized in detail and parts of them have caused us to make changes to the manuscript.

Below follows answers to the comments of the reviewers, and the detailed descriptions of action we have taken to each of them. In this letter the pages and line numbers refer to the revised manuscript. All the changes in the manuscript caused by reviewers' comments are typed in red.

Responds to the Reviewers' comments:

**Reviewer 1**

1) Zhou et al. present a system for laser stabilization based on NICE-OHMS. Instead of directly feeding back to the laser, they use a single-sideband modulator as a frequency actuator. This increases the flexibility of the system, as demonstrated by the similar performance achieved with two different lasers. While the system is well designed and implemented, I'm not sure that the results presented relevant enough to the atmospheric community or novel enough for publication in AMT.

   **Answer:** Thanks to the reviewer's comment. NICE-OHMS has been used to the fields of metrology, trace gas detection, ion detection and molecular spectroscopy since it has been invented. This manuscript is, for the first time, the intention to apply this technique to the LIDAR-based air monitoring. For the LIDAR system, such as for the detection of $CO_2$ in the upper atmosphere, the laser frequency should be exactly locked to the center of target transition and kept stability in short-term and long-term to ensure an accurate measurement. NICE-OHMS is a good candidate for this purpose due to its unique advantages, such as the properties of ultra sensitive spectroscopy measurement and detection ability of sub-Doppler spectrum. More

[Figure]

Shanxi University, 030006 Taiyuan, P.R.China
State Key Laboratory of Quantum Optics and
Quantum Optics Devices
Institute of Laser Spectroscopy
Weiguang Ma
Phone: +86 (0)13834595365
Fax: +86 351 7018227
E-mail: mwg@sxu.edu.cn
http://laserspec.sxu.edu.cn/English/HOME.htm

importantly, the utilization of single sideband modulator (SSB) makes sub-Doppler NICE-OHMS system more universal and compatible with different type of lasers. Therefore we think the involved works open a new vision to the atmospheric community and novel enough for the publication in AMT.

2) For the applicability to the atmospheric community, more details need to be provided about the potential application to LIDAR:

• First, what frequency stability is required for LIDAR and why?

**Answer:** Thanks to the reviewer's comments. In ideality, the better the frequency stabilization of laser frequency, the more accuracy of the result of LIDAR. In reality, the required frequency stability is ranging from hundreds of kHz to dozens of MHz, which depends on the linewidth of the target gas spectrum [1]. For example, Jet Propulsion Laboratory in USA has designed a Lidar system with frequency error of ~2 MHz for detection of $v_1+v_3$ band of acetylene [2]; The group in University of Maryland designed a system with frequency stability better than 0.3 MHz for the detection of $CO_2$ at the wavelength of 1572 nm [3]. All these information is inserted into the first paragraph of introduction.

• My initial thought is that NICE-OHMS is overly complicated for this application. The motivation for using NICE-OHMS was not well justified. What performance has been achieved with a wavemeter and with other schemes such as WMS/FM spectroscopy in a cavity? Why is NICE-OHMS necessary?

**Answer:** Thanks to the reviewer's comments. Benefiting from the modularized fiber laser and integrated fiber components, NICE-OHMS instrument is getting more and more compact than its inception. On the other hand, molecular transitions are the best frequency reference especially for the long-term laser stability since the frequency accuracy of a wavelength meter is normally in the order of tens MHz and meanwhile strongly relies on the ambient working conditions. In this paper we suggested a sub-Doppler spectrum, which is more sensitive to frequency excursion from transition center than Doppler broadened spectrum, to be applied to the high-performance frequency stabilization.

For the LIDAR-based air monitoring system, the absorption path length is normally in the order of tens kilometers. A weak transition for the target gas has to be selected to ensure the laser power can not be fully absorbed in the measuring height range. In order to get a frequency stabilized laser, a highly sensitive laser absorption spectroscopy technique should be employed to get a high SNR error signal for laser frequency locking. As you suggested that a cavity can be used to perform a high sensitive measurement together with the techniques of WMS and FMS. However,

WMS can only be used with the OA-ICOS or other cavity enhanced techniques without frequency locking but with a relatively low signal to noise ratio [4]. By the way, NICE-OHMS, also named as cavity enhanced frequency modulation spectroscopy, is just based on the combination of FMS and cavity enhanced absorption spectroscopy, which can obtain ultra sensitivity and high intracavity buildup power to get sub-Doppler signal of weak transition.

• What gas is expected to be used for the LIDAR application ($C_2H_2$ is not the most atmospherically relevant)? How will the performance compare (i.e., what are the linestrengths, expected linewidths, etc.)?

> **Answer:** Thanks to the reviewer's comment. I agree $C_2H_2$ is not the most atmospherically relevant. In the LIDAR-based gas monitoring applications, $CO_2$, $CH_4$, $N_2O$ et al. in the atmosphere are normally monitored [Applied Physics B Vol. 90, 593–608 (2008)]. In this submitted paper, $C_2H_2$ is used to demonstrate the feasibility of our NICE-OHMS system, just like the works in Jet Propulsion Laboratory [2]. As we know that the amplitude of saturation spectroscopy relies on the pump laser power and dipole moment of target transition. The dipole moment can be calculated by use of linestrength, partition function, life time and so on. The linewidth of the obtained sub Doppler saturation spectroscopy is normally in the order of several MHz.

• For the novelty, the included Gatti et al. reference has already demonstrated a wide bandwidth PDH lock using a single sideband modulator. This fact should be clarified in the current paper and the differences between this and the Gatti work should be discussed (e.g., the feedback bandwidth was higher in Gatti et al). Also, why was a lower bandwidth VCO used for VCO1? Demonstrating the ability to use a DFB or ECDL would make this a stronger paper.

> **Answer:** Thanks to the reviewer's comment and suggestion. The published paper by D. Gatti et al. do provide a good suggestion to the frequency locking of laser to an external cavity based on a single sideband modulator. However NICE-OHMS is a systematic work, which includes not only a technique of frequency locking, but also other techniques and methods such as DeVoe-Brewer locking, frequency modulation spectroscopy and spectrum analysis. The paper's novelty is obvious. Actually, this paper extends the application of the technique suggested by D. Gatti et al. to the NICE-OHMS. We will change the sentence "It would be possible to increase the bandwidth of this PDH servo to 3 MHz if a high bandwidth VCO would be employed (Gatti et al., 2015)." in the original version to "As suggested by D. Gatti, et al., the bandwidth of PDH servo can be increased to 5 MHz if a higher

bandwidth VCO and a good PID servo would be employed (Gatti et al., 2015)" in line 13 of page 5 in the revised version. Actually, we are devoting ourselves to find a VCO with narrow linewidth and wide bandwidth right now. Definitely, a NICE-OHMS system based on DFB or ECDL by use of a single sideband modulator with wide bandwidth PDH locking would be considered as a technique improvement.

3) Finally, I have some more specific comments/questions as well:

• "DVB" and "F-P cavity" are not defined in the caption to Figure 1

**Answer:** Thanks for the reviewer's suggestion. The corrections are done in the revised version.

• Why were OAPs used instead of lenses?

**Answer:** Thanks to the reviewer's comment. The substitution of lenses by OAPs is aimed to relieve the generation of etalon noise, which is caused by the reflection between two optical surfaces. The previous work has proved its validity [5].

• For the LIDAR output, will the frequency need to be tuned, e.g., to do an online/ off-line measurement? If so, how will this be accomplished?

**Answer:** For a differential absorption LIDAR, the wavelength should be switched on or off target gas resonance[1]. However the technique introduced in this paper is aimed to the gas measurement with fixed optical frequency.

• For Figure 2, it would be better to give the noise PSD. Also, I assume that this is an in-loop measurement? This should be clarified. In addition, I believe that this should be corrected for the cavity low-pass filter effect (e.g., Fig 5 in Gatti et al)?

**Answer:** Thanks for the comments and suggestions. The noise power spectrum is measured by a electronic spectrum analyzer under the in-loop condition. In this submitted paper, the figure 2 is mainly used to demonstrate the similarity of our system for the two applied lasers by showing its robust restrain of frequency noise in low frequency region, the servo bandwidth of 200 kHz and its universality. As a consequence, the modification of y-scale is not significant. In order to more clearly describe the presented noise properties, "Both the Video Bandwidth (VBW) and Resolution Bandwidth (RBW) in the measurement are 30Hz." is inserted to line 6 of page 5. On the other hand, the low pass filter effect should be used to correct the noise power spectrum for the analysis of frequency derivations at different Fourier frequencies. In this paper, we don't care so much on this point due to the noise immune properties.

[Figure]

Therefore, "Figure 2 shows frequency noise spectra of the error signals for PDH locking." is changed to "Figure 2 shows the frequency noise spectra of the in-loop error signals for PDH locking, which is measured by a electronical spectrum analyzer (FSW, Rohde&Schwarz, Germany)." in line 3 of page 5.

• What causes the upturn in the Allan deviation past 200 seconds?

**Answer:** Thanks to the reviewer's comment. "The upturn of Allan deviation is due to the residual amplitude modulation of fiber EOM, the not perfect design of PDH servo and etalon noise in the beam path." is inserted to line 23 of page 6.

• What was the temperature stability of the lab? How would the performance be affected by reduced temperature stability?

**Answer:** Thanks to the reviewer's comment. The lab is not temperature controlled and strongly depends on the air temperature outside the lab. Meanwhile several heaters are equipped in the lab in winter. Generally, the room temperature can change 0.5 K/hour. "During the measurement, the averaged temperature stability of the cavity is better than 0.5 K/hour since the cavity is exposed in the air." is inserted into line 15 of page 4. The reduced temperature stability will absolutely improve the long-term performance of the system, because the residual amplitude modulation of fiber EOM and etalon noise in the beam path are easily influenced by the temperature.

**Reviewer 2**

1) This paper develops a NICE-OHMS-based laser stabilisation system for potential application for lidar. The authors provide a brief overview of the relevant technical aspects of the paper, including sub-Doppler spectroscopy, the NICE-OHMS technique and the laser technology used in various NICE-OHMS demonstrations. The highlight of the paper is the application and use of a fiber-coupled optical single-sideband electooptic modulator (f-SSM) to make the system useful for laser stabilisation to a variety of lasers, and they demonstrated this by stabilising both an Erbium-doped fiber laser (EDFL) and a whispering gallery mode (WGM) laser. After producing a sD NICEC1 OHMS signal with each laser (using acetylene as a reference gas), each laser was then stabilised to that acetylene transition frequency using the sD NICE-OHMS signal produced. The stability of the stabilised output was measured and found to exceed the level needed for use in lidar systems.

The paper is well organised and clearly written, for the most part. Although the applicability of this system to lidar needs to be expanded upon, I see no reason why it

could not be used as a stabilized seed laser for lidar.

**Answer:** Thanks for your positive comments.

**Specific comments:**

2)  What was the acetylene gas source for your initial experimental realization of this, and how did you measure that it was at 100mTorr? Do you know the long-term stability of the pressure in the system? Would you expect to use the same system setup "in the field" when supporting a Lidar system?

    **Answer:** Thanks for the comments. "total pressure and 1000 ppm concentration" and "The pressure leaking rate for the cavity was smaller than 0.01 mTorr/min after it was closed by the vacuum valve (Leycon 215379, Oerlikon, Germany)." are inserted into the line 18 of page 5. The sub-Doppler NICE-OHMS setup should be integrated and miniaturized before it is applied to the Lidar system. In this status, it can not be applied to the Lidar system.

3)  Can you be more explicit about the stability requirements of a lidar system? What about the power required to stabilize such a system, can you give the details of the power requirement and can your system provide that power?

    **Answer:** Thanks for the comments. "In reality, the required frequency stability is ranging from hundreds of kHz to dozens of MHz, which relies on the linewidth of the target gas spectrum (Ehret et al., 2008)." and "Therefore, the Jet Propulsion Laboratory in USA has designed a frequency stabilization system for coherent Lidar application with frequency error of 2.06 MHz for detection of $v_1+v_3$ band of acetylene (Meras et al., 2008); The group in University of Maryland has designed another frequency stabilization system with frequency drift better than 0.3 MHz for the detection of $CO_2$ at the wavelength of 1572 nm (Numata et al., 2011)." are insert to line 31 of page 1.

4)  Lidar systems also would like to target other molecules, especially those related to greenhouse gas emissions ($CO_2$, methane, etc.) Can you comment on how adaptable your system is to these other molecular species?

    **Answer:** Thanks for the comments. "On the other hand, if this setup is applied to the detection of $CO_2$, $CH_4$ or other gases, the saturation power of the target molecular transition, as a function of dipole moment, laser beam waist and total pressure, should be determined firstly to choose a suitable laser and a proper cavity design. The transition dipole moment is the function of its linestrength, life time and partition function (Ma et al., 2008). Normally, for a weak absorption line with the saturation power in the order of hundreds of watts, a cavity with finesse of more

than tens thousands and a laser with its linewidth of the order of kHz are required., a cavity with finesse of more than tens thousands and a laser with its linewidth of the order of kHz are required." is added to line 14 of page 7. "The saturation power under the pressure of 100 mTorr was around 700 mW (Foltynowicz et al., 2008b)" is inserted to line 23 of page 5. "The corresponding saturation power under the pressure of 100 mTorr was around 800mW." is inserted to line 32 of page 5.

5) Can you provide any comments at the end of your paper to indicate what changes would need to be made to your system for it to work reliably outside a research laboratory, "in the field"?

    **Answer:** Thanks for the comments. "Although the frequency stability can satisfy the requirements of a real Lidar system, the whole setup should be miniaturized and a seismic design is necessary before it is applied to the real field." is inserted into line 12 of page 7.

6) Page 3: Line 1: I am confused by this first sentence. The "noise immunity" obviously refers to "noise" sources - not the actual signal. The "background signal" you mention is induced by other processes unrelated to the "noise" in the "noise immunity" – from optical power variations in the incident light to the cavity, or coupling efficiency. Can you please clarify or remove this sentence?

    **Answer:** Thanks for the comments. The noise in "noise immunity" refers solely to the frequency to amplitude noise. The influence of this noise to the NICE-OHMS signal (i.e. the demodulated cavity enhanced frequency modulation signal) is null only under the condition of empty cavity, zero residual amplitude modulation from EOM and zero other background noise before the cavity. If the laser addresses the molecular transition, the absorption will make the cavity transmitted triplet different noise level and therefore destroy the noise immunity property to some extend. We remove "often referred to as a background signal" from the line in the text.

7) Page 6: Line 6: "for integration times up to 240 sec" - I might argue that you have overestimated your white noise response window, especially in the case of the fiber laser Allan deviation data (black points and curve). I would estimate (reading directly off Figure 4) that there is a white noise response out to 100-120 seconds (the intersection of your "white noise" line and a flat line indicating the ADEV value when the ADEV begins drifting up), rather than 240 seconds, for the fiber laser. By 240 seconds you are clearly out of the white noise regime. However, it is a bit difficult to see this clearly from the graph. Additionally, it is not obvious that there are any "dotted" lines on this plot, as mentioned in the figure caption - the "white noise response" lines appear solid. Figure 3: To be able to better compare the signal-to-noise-ratio of the sD

NICE-OHMS signals in sub-plots (a) and (b), it would be helpful to have the y-axes have the same minimum and maximum values (perhaps 0.7 or 0.65).

**Answer:** Thanks for your reminding. The corresponding changes have done in the revised version.

8)  Technical corrections:

• Page 1: Line 1: Typically one would use "a universal" rather than "an universal". Line 26: "a relative weak" -> a relatively weak;

• Page 2: Line 22: "cavity with a finesse of 105" - > "cavity with a finesse of 100,000" or "cavity with a finesse of 10^5" Line 24: should be rewritten - "sensitivity of 1x10^-14 cm-1 at 1-s averaging time."

• Page 3: Line 6/7: "more than one frequency actuators are usually utilized often a" - >"more than one frequency actuator is usually utilized, often a"

• Page 5: Line 30: "The performance of the this frequency stabilization were assessed by" -> "The performance of the this frequency stabilization were was assessed by" Page 6: Line 15: "testified" - > "tested"

**Answer:** The corrections are done in the revised version.

9)  Generally there is sometimes an issue of spacing between text and the beginning or end of parentheses that should be checked for throughout the paper.

**Answer:** The corrections are done in the revised version.

10) It is a bit confusing to have two very similar References (Ehlers, 2014) and (Ehlers et al., 2014) in the paper text, there a way to differentiate? (Ehlers [Thesis], 2014)? Also, I am not sure that the page 7, line 24 Ehlers reference, is correct. Perhaps use "PhD dissertation" rather than "Doctor". "Ehlers, P.: Further development of NICE-OHMS – an ultra-sensitive frequency-modulated cavity-enhanced laser-based spectroscopic technique for detection of molecules in gas phase, PhD dissertation, Umeå universitet, Umeå, 2014."

**Answer:** The corrections for all the references are done in the revised version.

**Reviewer 3**

1)  The manuscript tilted by "Laser frequency stabilization based on an universal sub-Doppler NICE-OHMS instrumentation for the potential application in atmospheric Lidar" made a frequency stabilized laser based on cavity enhanced optical heterodyne molecular spectroscopy named as NICE-OHMS.

The text line of 30 in the page 5 is as following: "The performance of the frequency

stabilization were assessed by Allan plot of the frequency deviation estimated from the error signals, i.e. the sD NICE-OHMS signals, calibrated by the slopes at the zero crossing point of the sD signals."

Analysis of servo error signal is not right way to characterize the performance of frequency stabilized laser. The electronic servo box can drive the laser frequency to make error signal at the zero-crossing point of the sD signals. Fig.3 shows the baseline of sD signals (blue line) is moving around the zero-crossing point. The servo box adjusted the laser frequency to make error signal at the zero-crossing point, therefore the laser frequency is unstable. So I hope the authors to make two independent frequency stabilized lasers and make analysis of the beating signal between two lasers to characterize the performance of frequency stabilized laser.

> **Answer:** Thanks for your suggestions on the paper. For evaluating the frequency stability of a stabilized laser, we agree that the close-loop frequency deviation as a function of time can be determined by the beat signal of two duplicate systems, however, as a alternative, it can also be determined by the calibrated close-loop error signal [6, 7]. For simplicity, our paper follows the latter method.
>
> Therefore, there are no revisions on the paper for up-mentioned comments.

2) The performance of stabilized laser is dependent on the length of cavity, pressure of cell, input light power to the cavity, beam size inside of the cavity, cell temperature instability, the cavity output light power and the RAM in the sD signal. So the authors need to add analysis and optimization for the parameters which affect the frequency instability of the stabilized lasers.

> **Answer:** Thanks for the comments. For the sub Doppler NICE-OHMS signal, as you mentioned, the gas pressure, intracavity power and the beam size inside the cavity will influence the amplitude and linwidth of the sub-Doppler signal, which definitely causes different laser locking performance[8, 9]. However the aim of this paper is to show that the sub Doppler NICE-OHMS signal can be used to the frequency stabilization in the field of atmospheric LIDAR even under the not fully optimized conditions. Therefore "However the laser frequency locking performance is strongly influenced by the amplitude and linewidth of sD NICE-OHMS signal which will be changed by the gas pressure and intracavtiy power. Although the locking results in the paper have satisfied the requirements of atmospheric LIDAR applications, if the NICE-OHMS system is optimized, an even better result can be expected." is inserted after the last paragraph of section 4.
>
> The residual amplitude modulation (RAM) will cause the drift of baseline of the sD signal. In order to suppress the RAM, a fiber EOM with proton exchanged

waveguide is used in this system since this type of EOM can filter out the polarization component along the ordinary axis. More detail can read the reference [10]. "The upturn of Allan deviation is due to the not fully suppressed residual amplitude modulation of fiber EOM, the not perfect design of PDH servo and etalon noise in the beam path." is inserted into the last of second paragraph of section 4.

"During the measurement, the averaged temperature stability of the cavity is better than 0.5 K/hour since the cavity is exposed in the air." is inserted to the cavity description of the paper. The variation of the cavity temperature will cause different Doppler broadening linewidth, however the sub-Doppler signal is obtained by addressing the molecules with zero velocity component along the beam direction. As a result, the temperature will not evidently influence the error signal.

When the NICE-OHMS is performed, the cavity length is locked to the laser frequency. Meanwhile, the sD signal is obtained by scanning the cavity length thereby the laser frequency. Therefore the cavity length will not influence the error signal.

A new sD NICE-OHMS system based on a f-SSM and 100000 finesse cavity is being performed in our lab. All the up-mentioned factors will be considered in our next works.

References

1.   G. Ehret, C. Kiemle, M. Wirth, A. Amediek, A. Fix, and S. Houweling, "Space-borne remote sensing of $CO_2$, $CH_4$, and $N_2O$ by integrated path differential absorption lidar: a sensitivity analysis," Appl. Phys. B. **90**, 593-608 (2008).

2.   P. M. Jr., I. Y. Poberezhskiy, D. H. Chang, J. Levin, and G. D. Spiers, http://citeseerx.ist.psu.edu/viewdoc/download?doi=10.1.1.546.4790&rep=rep1&type=pdf.

3.   K. Numata, J. R. Chen, S. T. Wu, J. B. Abshire, and M. A. Krainak, "Frequency stabilization of distributed-feedback laser diodes at 1572 nm for lidar measurements of atmospheric carbon dioxide," Appl. Optics. **50**, 1047-1056 (2011).

4.   V. L. Kasyutich, C. E. Canosa-Mas, C. Pfrang, S. Vaughan, and R. P. Wayne, "Off-axis continuous-wave cavity-enhanced absorption spectroscopy of narrow-band and broadband absorbers using red diode lasers," Appl. Phys. B. **75**, 755-761 (2002).

5.   I. Silander, T. Hausmaninger, and O. Axner, "Model for in-coupling of etalons into signal strengths extracted from spectral line shape fitting and methodology for predicting the optimum scanning range-demonstration of Doppler-broadened, noise-immune, cavity-enhanced optical heterodyne molecular spectroscopy down to 9 x 10(-14) cm(-1)," Journal of the Optical Society of America B-Optical Physics **32**, 2104-2114 (2015).

6.   T. L. Chen and Y. W. Liu, "Noise-immune cavity-enhanced optical heterodyne molecular spectrometry on $N_2O$ 1.283 $\mu$m transition based on a quantum-dot external-cavity diode laser," Optics Letters **40**, 4352-4355 (2015).

[Figure]

7.      H. Dinesan, E. Fasci, A. Castrillo, and L. Gianfrani, "Absolute frequency stabilization of an extended-cavity diode laser by means of noise-immune cavity-enhanced optical heterodyne molecular spectroscopy," Optics Letters **39**, 2198-2201 (2014).

8.      O. Axner, W. Ma, and A. Foltynowicz, "Sub-Doppler dispersion and noise-immune cavity-enhanced optical heterodyne molecular spectroscopy revised," Journal of the Optical Society of America B-Optical Physics **25**, 1166-1177 (2008).

9.      A. Foltynowicz, W. Ma, and O. Axner, "Characterization of fiber-laser-based sub-Doppler NICE-OHMS for quantitative trace gas detection," Opt. Express. **16**, 14689-14702 (2008).

10.     I. Silander, P. Ehlers, J. Wang, and O. Axner, "Frequency modulation background signals from fiber-based electro optic modulators are caused by crosstalk," Journal of the Optical Society of America B-Optical Physics **29**, 916-923 (2012).

Sincerely yours

Weiguang Ma

State Key Laboratory of Quantum Optics and Quantum Optics Devices
Institute of Laser Spectroscopy
Shanxi University
Taiyuan, P. R. China
Email: mwg@sxu.edu.cn
Tel: +86 – (0) 13834595365
FAX: +86 – (0) 351 7018227